# Intakes of Total, Free, and Naturally Occurring Sugars in the French-Speaking Adult Population of the Province of Québec, Canada: The PREDISE Study

**DOI:** 10.3390/nu11102317

**Published:** 2019-09-30

**Authors:** Amélie Bergeron, Marie-Ève Labonté, Didier Brassard, Alexandra Bédard, Catherine Laramée, Julie Robitaille, Sophie Desroches, Véronique Provencher, Charles Couillard, Marie-Claude Vohl, Benoît Lamarche, Simone Lemieux

**Affiliations:** 1Institute of Nutrition and Functional Foods, Université Laval, Québec, QC G1V 0A6, Canada; Amelie.Bergeron.6@ulaval.ca (A.B.); Marie-Eve.Labonte@fsaa.ulaval.ca (M.-È.L.); Didier.Brassard.1@ulaval.ca (D.B.); alexandra.bedard.1@ulaval.ca (A.B.); Catherine.Laramee@fsaa.ulaval.ca (C.L.); Julie.Robitaille@fsaa.ulaval.ca (J.R.); Sophie.Desroches@fsaa.ulaval.ca (S.D.); Veronique.Provencher@fsaa.ulaval.ca (V.P.); Charles.Couillard@fsaa.ulaval.ca (C.C.); Marie-Claude.Vohl@fsaa.ulaval.ca (M.-C.V.); Benoit.Lamarche@fsaa.ulaval.ca (B.L.); 2School of Nutrition, Université Laval, Québec, QC G1V 0A6, Canada

**Keywords:** nutrition, sugar intakes, free sugars, naturally occurring sugars, 24-h recall, Québec, Canada, PREDISE study

## Abstract

The objective of this study was to characterize the intakes of different types of sugars in an age- and sex-representative sample of French-speaking adults from five regions of the Province of Québec, Canada, enrolled in the cross-sectional PREDISE (PRÉDicteurs Individuels, Sociaux et Environnementaux) study (*n* = 1147, 18–65 years old; 50.2% women). Because only total sugar content of foods and beverages is available in the Canadian Nutrient File (CNF) 2015, the initial step of this study was thus to build a database of free and naturally occurring sugars content of each food item and recipe included in the R24W, which is an automated, self-administered, web-based, 24-h dietary recall validated to estimate nutrient intakes in French-speaking adults of the Province of Québec. Total sugars were manually differentiated into free and naturally occurring sugars using a systematic algorithm based on previously published systematic algorithms. The World Health Organization (WHO)’s free sugar definition was used to differentiate total sugars into free and naturally occurring sugars. Dietary intake estimates were assessed using three 24-h dietary recalls completed with the R24W. Mean total, free, and naturally occurring sugar intakes were 116.4 g (19.3% of daily energy intake (%E)), 72.5 g (11.7%E), and 44.0 g (7.5%E), respectively. Over half (57.3%) of the overall sample did not meet the WHO’s recommendation to consume less than 10%E from free sugars. Women had a higher %E from naturally occurring sugars than men and being younger was associated with a greater %E from free sugars. Sugar intakes among French-speaking adults from the Province of Québec were mainly in the form of free sugars, with the majority of the population exceeding the WHO recommendation regarding free sugar intake. This suggests that public health efforts towards reducing free sugar intake in this population are relevant and necessary, considering that overconsumption of free sugars negatively influences health outcomes.

## 1. Introduction

Sugar intake is a worldwide concern, a public health issue, and an active area of research because of its effect on excessive caloric intake and body weight gain as well as on health outcomes [1]. Some food sources of sugars, such as fruits and vegetables, in which sugars are naturally occurring, are known to have beneficial health effects [2,3,4], whereas other sources, such as sugar-sweetened beverages (SSBs), containing free sugars, are deleterious to health [5,6]. A distinction between the different types of sugars (i.e., total, free, and naturally occurring sugars) is thus crucial to best appreciate the association between sugar intake and health [7].

Many countries and health organizations have set qualitative or quantitative recommendations regarding total, free, or added sugar intake [8] The World Health Organization (WHO) recommends limiting free sugar intake to a maximum of 10% of daily energy (%E) and even proposes to reduce free sugar intake below 5%E, as a conditional recommendation [9]. Health Canada, through its 2019 Canada’s Food Guide, recommends limiting free sugar intake by reducing consumption of highly processed foods and by consuming nutritious food containing little to no free sugars [10]. National data from various Western countries have shown that a vast majority of populations do not reach the WHO’s target [11,12,13,14]. However, data on free and naturally occurring sugar intakes are utterly lacking in Canada because only total sugar content of foods is available in the Canadian Nutrient File 2015 [15,16]. One reason explaining why nutrient databases in Canada and other countries usually contain information only on total sugar [14] is that free and naturally occurring sugars are chemically indistinguishable [17]. Accordingly, the Canadian Community Health Survey (CCHS), which relies on the Canadian Nutrient File (CNF) 2015, only provides information on total sugar intake [16]. 

The lack of information about the different types of sugars in nutrient databases and the fact that food manufacturers are not keen to provide this information on the nutrition facts table explain why we need to rely on systematic methods to estimate free and naturally occurring sugar content of foods. Louie et al. [18] proposed a step-by-step algorithm to differentiate free and naturally occurring sugars from a nutritional database through an individual methodical reflection for each food item. It is currently the best possible sugar differentiation approach, in comparison with other less precise methodologies such as, for example, the use of market availability data with corrections for wastage [19]. Furthermore, such a systematic approach applied on individual consumption data allow the use of the results beyond the sole purpose of portraying population-level intakes. In fact, these systematic approaches also make it possible to assess the associations between sugar intakes and other variables of interest such as cardiometabolic risk factors.

The objective of this study was to characterize and describe free and naturally occurring sugar intakes in an age- and sex-representative sample of French-speaking adults from five regions of the Province of Québec, Canada. It is the first time that such a study with a systematic sugar differentiation method was conducted in Canada. This has been done by first differentiating types of sugars for each food item comprised in a food composition database based on the CNF 2015 and integrated into the R24W, a web-based 24-h dietary recall, which is used to estimate the dietary intakes of the study’s participants. Our hypotheses are that the majority of study participants do not meet WHO’s recommendation to consume less than 10%E from free sugars and that sugar intakes of adults from the Province of Québec are comparable to population intakes in other Western countries. 

## 2. Materials and Methods 

### 2.1. Participants and Procedures

The PREDISE (PRÉDicteurs Individuels, Sociaux et Environnementaux) study was designed to assess how individual, social, and environmental factors are associated with adherence to dietary guidelines in force at the time of study’s realization, which were set out in the 2007 Canada’s Food Guide [20]. The PREDISE study design and methodology were previously detailed elsewhere [21]. Briefly, the study had a web-based, multicentered, cross-sectional design. Between August 2015 and April 2017, recruitment was done using random digit dialing in five different administrative regions from the Province of Québec, Canada: Capitale-Nationale/Chaudière-Appalaches, Estrie, Mauricie, Montréal, and Saguenay-Lac-St-Jean. A priori recruitment quotas were defined to obtain representativeness in each of the five regions upon sex and predetermined age groups (18–34, 35–49, and 50–65 years). Participants completed online socio-demographic questionnaires and three web-based 24-h dietary recalls using the R24W (described below). Once the online part was completed, participants were invited to their region’s research center for anthropometric measurements and blood sampling. The final study sample included 1147 participants (50.2% women). Participants were mostly Caucasian (94.3%) with a university degree (44.5%) and 94.6% of them completed three web-based 24-h dietary recalls [21]. Written informed consent from all participants was obtained. The project was conducted in accordance with the Declaration of Helsinki and was approved by the Research Ethics Committees of Université Laval (ethics number: 2014-271), Centre hospitalier universitaire de Sherbrooke (ethics number: MP-31-2015-997), Montreal Clinical Research Institute (ethics number: 2015-02), and Université du Québec à Trois-Rivières (ethics number: 15-2009-07.13).

### 2.2. R24W Database

The PREDISE study used an automated, self-administered, validated [22,23,24], web-based tool, the R24W, to obtain 24-h dietary recalls. The R24W database includes nutritional values of foods that participants can select while completing their 24-h dietary recalls. It was developed by our research team [25], which allows us to improve and modify the tool autonomously. It includes 2669 food items (1203 items refer to manually created recipes [26]) whose nutritional values are predominantly linked to the Canadian Nutrient File (CNF) 2015, the most recent version available [15]. For a few food items, when the food was not available in the CNF, nutritional values were based on the United States Department of Agriculture Nutrient Database for Standard Reference [27] (5.8% of total items). 

There are some missing values for total sugars in the CNF 2015. The percentage of coverage for total sugars in the CNF 2015 is 81.6% [28]. The percentage of coverage is the percentage of foods for which a value of the indicated nutrient is available. Some of CNF 2015 foods, but not all, were selected to constitute the R24W database, for which the percentage of coverage for total sugars was 95.7%. Therefore, for the 115 foods with missing total sugar value and the 133 recipes having one or more ingredients with missing total sugar content (representing 248 food items; 9% of all food items in the R24W database), a value was assigned. For missing total sugar content of foods or recipe ingredients, values were mostly assigned by proportionally comparing their carbohydrate content with carbohydrate and total sugar contents of similar foods. When it was not possible to use the latter approach, we assigned a total sugar value by consulting the nutrition facts table of a similar commercial product or by referring to the Food Label Information Program (FLIP) database (2013 version), which is a Canadian database comprising the nutritional values of over 15,000 commercial foods and drinks, including information about the content of free and naturally occurring sugars [29]. 

### 2.3. Sugar Differentiation

As indicated above, only total sugar content is available in the CNF 2015, and thus in the R24W database. Free and naturally occurring sugars were manually differentiated in the R24W database’s foods and mixed dishes. The WHO’s free sugar definition was used to differentiate total sugars into free and naturally occurring sugars: “Free sugars include monosaccharides and disaccharides added to foods and beverages by the manufacturer, cook or consumer, and sugars naturally present in honey, syrups, fruit juices and fruit juice concentrates” [9]. Naturally occurring sugars are, by definition, intrinsically present in foods, such as whole fruits and vegetables, and unsweetened milk. The sugar differentiation method used was adapted from the algorithm proposed by Bernstein et al. [29] (which is itself adapted from Louie et al.’s [18]). We chose not to directly use the algorithm from Louie et al. because one of the steps of the algorithm uses specific sugar content (i.e., lactose and maltose) and this information was not available for all food items in our database. Also, the algorithm of Louie et al. has been developed to estimate added sugars, while we wanted to obtain free sugar information. To better adapt the algorithm from Bernstein et al. [29] to our database, we modified one of the steps and added two new steps. Our method allows to methodically differentiate free sugars from naturally occurring sugars of every food item in our R24W database, step by step. Table 1 shows the different steps of the modified algorithm that we used. After differentiating total sugars into free and naturally occurring sugars, we assessed the mean consumption of total, free, and naturally occurring sugars for each study participant and for the overall sample, based on three web-based 24-h recalls. 

### 2.4. Plausibility of Self-Reported Energy Intakes

The plausibility of self-reported energy intakes was determined with a method described by Huang et al. [30] comparing reported energy intake to predicted energy requirements, as calculated using the Institute of Medicine equations [31]. Participants with a ratio of self-reported energy intake over predicted energy requirements <0.78 or >1.22 were considered under-reporters and over-reporters, respectively. Details of the methodology used to assess the plausibility of self-reported energy intakes are presented elsewhere [32].

### 2.5. Statistical Analyses

Analyses were performed using SURVEY procedures in SAS Studio (version 3.6), accounting for the stratified design of the PREDISE study. Sampling weights were calculated to balance the sample size, which was larger than anticipated at the elaboration stage of the study. Missing sociodemographic characteristics (body mass index (BMI) (*n* = 125), education (*n* = 60), and income (*n* = 60) levels) were imputed using the fully efficient fractional imputation method, as presented elsewhere [21]. To determine the plausibility of self-reported energy intakes in participants who had missing data for height and weight, a single imputation using the hot deck method was performed (*n* = 125). Multivariable linear regressions were performed to obtain least square means of dietary intakes of sociodemographic subgroups with Tukey’s post hoc tests to assess potential differences. Sex, age groups (18–34 years, 35–49 years, 50–65 years), administrative region (Capitale-Nationale/Chaudière-Appalaches, Estrie, Mauricie, Montréal, and Saguenay-Lac-St-Jean), BMI groups (normal <25.0 kg/m^2^, overweight 25.0–29.9 kg/m^2^, obese ≥30.0 kg/m^2^), education level (high school or no diploma, CEGEP (CEGEP is a preuniversity and technical college institution particular to the Province of Québec educational system, which is considered higher than high school and lower than university), university), household income levels in Canadian dollars (<30,000, ≥30,000 to <60,000, ≥60,000 to <90,000, ≥90,000), reporting status (under-, plausible-, and over-reporters), and the number of 24-h dietary recalls completed on weekend days (0, 1, 2, 3) were used as covariates when appropriate. Frequency statistics were used to determine percentages of individuals in the overall sample and in sociodemographic subgroups meeting WHO’s recommendation to consume less than 10% of daily energy as free sugars. Chi-squared tests were conducted to assess differences between subgroups. Pearson correlations were performed to assess correlation coefficients between %E provided by total, free, and naturally occurring sugars. Considering the numerous statistical tests performed in the present study, *p* values lower than 0.001 were considered statistically significant.

## 3. Results

### 3.1. Mean Intakes of Total, Free, and Naturally Occurring Sugars

Table 2 shows mean intakes of total, free, and naturally occurring sugars, expressed in grams per day; percentages of total sugars; and the percentage of daily energy (%E) they provide. Mean values in the overall sample (i.e., “All”) are unadjusted, whereas values presented for the different subgroups are adjusted for sex, age group, administrative region, BMI, education, reporting status, household income level, and weekend, when appropriate. See Appendix A for unadjusted means.

In the overall sample, the mean intake of total sugars was 116.4 g, which represented 19.3% of daily kilocalories. Free sugars provided 11.7% of total kilocalories. Compared with men, women consumed less total sugars when expressed as absolute intakes in grams per day (*p* < 0.0001). Women also consumed less free sugars, but more naturally occurring sugars, than men when expressed as the proportion of total sugars. Also, the %E from naturally occurring sugars was greater in women than in men (*p* < 0.0001). Younger age was associated with a higher absolute intake of total sugars. A higher proportion of free sugars over total sugars and a lower proportion of naturally occurring sugars over total sugars were observed in younger participants when compared with older participants. Younger age was also associated with a higher %E from free sugars. Administrative region was not a factor influencing sugar intakes. As for the presence of obesity (BMI ≥ 30.0 kg/m^2^), it was only significantly associated with a higher absolute intake of total sugars. Each education group consumed about the same amount of total sugars when expressed in grams per day. However, holding a university degree was associated with a lower proportion of free sugars over total sugars and a higher %E from naturally occurring sugars. Under-reporters consumed the lowest absolute amount of total sugars, had the lowest proportion of total sugars being free sugars, and had the least %E from free sugars. The difference between under- and over-reporters was no longer statistically significant for total and naturally occurring sugars when expressed as %E. Household income and the number of food recalls completed on weekend days were not significantly associated with intakes of total, free, and naturally occurring sugars (data not shown).

### 3.2. Proportion of Individuals Meeting WHO’s Recommendation

Figure 1 shows that 42.7% of the overall sample met the WHO’s recommendation to consume less than 10%E as free sugars. The prevalence of participants meeting this recommendation did not significantly vary by sex, age, BMI, and education. However, the proportion of individuals meeting WHO’s recommendation differed significantly depending on the reporting status. In fact, 59.5% of under-reporters, 41.4% of plausible reporters, and 36.0% of over-reporters met the recommendation. Also, the prevalence of meeting the WHO’s recommendation did not vary by household income and by the number of food recalls completed on weekend days (data not shown).

### 3.3. Correlations among Types of Sugars

When considering the overall diet in the entire sample, energy from both free and naturally occurring sugars correlated positively with energy from total sugars, but the association was stronger with free sugars (*r* = 0.79; *p* < 0.0001) than with naturally occurring sugars (*r* = 0.45; *p* < 0.0001), as presented in Table 3. Energy from free sugars was inversely correlated with energy from naturally occurring sugars (*r* = −0.20; *p* < 0.0001). When looking at sugar intakes from foods only, %E from free sugars and %E from naturally occurring sugars similarly correlated with %E from total sugars (*r* = 0.69; *p* < 0.0001 and *r* = 0.66; *p* < 0.0001, respectively). %E from free sugars and %E from naturally occurring sugars were not significantly correlated (*r* = −0.09; *p* = 0.01). When considering drinks only, %E from free sugars correlated strongly with %E from total sugars (*r* = 0.91; *p* < 0.0001). In drinks, %E from naturally occurring sugars and from total sugars was also correlated (*r* = 0.39; *p* < 0.0001), while %E from naturally occurring sugars was not significantly correlated with %E from free sugars (*r* = −0.02; *p* = 0.61).

## 4. Discussion

To the best of our knowledge, this study is the first to describe the intakes of free and naturally occurring sugars of the adult population of the Province of Québec, based directly on reported food consumption and nutrient composition data. We found that the majority of total sugar intake comes from free sugars and that more than half of the French-speaking Québec adult population does not meet WHO’s recommendation to consume less than 10%E from free sugars.

The mean total sugar intake of the Québec adult population (19+ years) was reported in CCHS 2015 and was found to be about 20 g lower than the value obtained in the PREDISE sample [33]. As the prevalence of energy misreporting is lower in PREDISE (16.1%) than in CCHS 2015 (34.5%) [16,34], this may explain, in part, the lower value obtained in CCHS 2015. Another explanation for the difference observed between the two studies may be that, in CCHS 2015, the older subgroup (71 years+) had the lowest total sugar intake [33], an age group that was not represented in the PREDISE study. Of note, when total sugar intakes were expressed in %E, the value obtained in our study (i.e., 19.3%E) was exactly the same as that obtained in the Québec population of CCHS 2015 [33]. These results suggest that expressing sugar intakes as %E may be a better approach to make comparisons between studies, as it can reduce the bias associated with misreporting and can also reduce the influence of differences in energy intake between populations.

Of note, no other Canadian studies appear to have used a methodology similar to the one used here to assess intakes of free sugars at a population level. In a study from Langlois and Garriguet [35], the authors made the assumption that sugars from “other” foods (i.e., low nutritive value foods and beverages not included in any of the food categories of the 2007 Canada’s Food Guide [20] and whose consumption must be limited such as soft drinks, ice cream, candies, and chips) were unlikely to be naturally occurring sugars and they used the proportion of sugars from “other” foods on total sugars to provide an estimation of the proportion of free sugars over total sugars. The value obtained (35% of total sugars) [35] was much lower than what we found in the present study (59.6%). Indeed, free sugars are not solely found in “other” foods. For example, flavored yogurts contain free sugars, but were not considered as “other” foods in the previous analysis by Langlois et al. [35]. In another study in which the proportion of free sugars on total sugars in prepackaged foods in Canada in 2013 was calculated (using FLIP), the value obtained (62%) [29] was very similar to what we found in the present study. It is not surprising knowing that the sugar differentiation method we used shares similar steps with the algorithm used in FLIP [29], which is also the reference database used at a certain step of our differentiation method. Finally, although results from the Québec population of CCHS 2015 do not include data about free sugar intake (only data about total sugar intake were available), it was found that vegetables and fruits as well as milk and substitutes contributed more to total sugar intake in women than in men [33], which reflects that %E from naturally occurring sugars is probably higher in women, just as we found in our sample. Moreover, “other” foods contributed more to total sugar intake in men than in women [33], which is also consistent with our results.

Only a few studies from other countries have assessed free sugar intakes in their respective population [11,12,13,14,36,37,38,39]. All, except cross-sectional Swiss National Nutrition Survey 2014–2015 [11], used WHO’s free sugar definition. The %E from free sugars (11.7%) found in PREDISE was higher than what was observed in Spain (7.1%) [36], similar to results from Switzerland (11%) [11] and Australia (11.7%) [37] and lower than intakes reported in Germany (14%) [14]. To our knowledge, no population data about free sugar intake are available in the United States. However, the intake of added sugars was reported based on data from the NHANES survey (2015–2016). Intake of added sugars in adults was 16.2 teaspoon equivalents (68.2 g), which represented 13%E [40]. On the basis of the results from Kibblewhite and et al. [13], Sluik et al. [14], and Lei et al. [37], it can be suggested that free sugars usually provide 1%E–2%E more than added sugars. That would suggest that free sugar intake in the NHANES survey would be higher than what is observed in the present study (11.7%E).

Only a few studies have reported the intake of naturally occurring sugars. Ruiz et al. [36] reported a %E from naturally occurring sugars in Spain (9.3%E) higher than in the Québec population (7.5%E), whereas in Finland, the reported intake of naturally occurring sugars (6.3%E for women and 4.7%E for men) [41] was lower than in the Québec population. However, it should be noted that naturally occurring sugars were likely to be underestimated in the Finnish study because only sugars from fruits, vegetables, and fruit juices served to determine naturally occurring sugar intake [41].

Our results show that the majority of adults (55.4% of women and 59.2% of men) do not meet WHO’s recommendation on free sugar intake (i.e., less than 10%E from free sugars). This is also an observation made in other Western countries. In fact, our results are very similar to those obtained in Switzerland and New Zealand (56% of women and 55% of men [11] and 58% of the overall sample [13], respectively), slightly higher than what has been found in Australia (47%) [12], but lower than in the Dutch population (71% of women and 67% of men) [14]. However, the results from our study and from all of the studies mentioned are very different from data obtained in Japan, where only 13.3% of women and 8.7% of men consumed more than 10%E from free sugars [39]. It is not surprising knowing that, compared with the Western diet, the Japanese diet contains less confectionaries and SSBs [39].

In the present study, some sociodemographic factors were found to be associated with sugar intake. The contribution of naturally occurring sugars to energy intake was higher in women than in men, but unlike other results, the %E from total sugars was not higher in women than in men [14,41]. Just like in PREDISE, other studies have also shown that increasing age is associated with a lower intake of added [41,42,43] or free sugars [11,12]. On the basis of our results, being in the obese category is not significantly associated with the %E from free or naturally occurring sugars, while having a university degree is associated with a higher %E from naturally occurring sugars. These results are difficult to compare to previous studies because associations of BMI [41,42] and education [41,44] with sugar intakes are inconsistent in the literature. We also found in our study that the %E from free sugars was lower in under-reporters. To our knowledge, no other study has assessed free sugar intake according to the plausibility of reported energy intakes. Although some subgroups of our population were characterized by lower free sugar intake, none of them had a mean %E from free sugars lower than 10% (see Appendix A). Public health efforts to reach WHO’s target (i.e., less than 10%E from free sugars) are thus relevant and useful for all subgroups of our population.

As total sugar is often the only information available in nutrient databases and on food labels, we wanted to verify the strength of the correlations between total sugars and free and naturally occurring sugars. We found that in the overall diet (food and drinks together), both %E from free and naturally occurring sugars correlated positively with %E from total sugars, just like what was found in the study from Kaartinen et al. [41], for added and naturally occurring sugars. Our results showed that the association was stronger with free sugars than with naturally occurring sugars (*r* = 0.79 vs. *r* = 0.45), suggesting that total sugar intake more closely reflects free sugar intake than naturally occurring sugar intake in the overall diet. Also, we observed an inverse correlation between %E from free and naturally occurring sugars, which was also found by Kaartinen et al. [41] between added and naturally occurring sugars. This suggests that people eating more free sugars in proportion of daily energy also eat less naturally occurring sugars. When we analyzed drinks alone, for which the main sources of free sugars are fruit juices and soft drinks, we found that %E from free sugars correlated nearly perfectly with total sugar intake (*r* = 0.91). Therefore, on the basis of our results, focusing on total sugars on nutrition facts tables, as planned by the Government of Canada in its 2016 report on food labelling changes [45], seems to be an appropriate approach to guide Canadians to healthy choices, but mainly for drinks. The approach seems questionable for foods. Indeed, total and free sugars in drinks correlated more strongly than total and free sugars in foods (*r* = 0.69). Adding directly the free sugar content on nutrition facts tables, for both foods and drinks, would better inform the consumer.

### Strengths and Limits

The principal strengths of this study lie in the survey methodology itself. Participants were recruited with random digit-dialing in order to be representative in terms of age and sex of the adult French-speaking population of five of the main administrative regions of the Province of Québec. They completed three self-administered, web-based, validated, 24-h dietary recalls on randomly allocated days. The large manually created recipes database used in the R24W also makes it unique. Free and naturally occurring sugars of mixed dishes are then precisely differentiated. On the basis of the current literature, we consider that we used the best possible approach to accurately differentiate total sugars in free and naturally occurring sugars. The method is simple, but it has to be emphasized that it is a time-consuming process.

Our study also has limitations. Just like every other similar method, subjectivity of certain steps may possibly affect the estimation of free sugar content. A total of 16.5% of individual food items from our database was concerned by steps involving subjective decisions. Despite being a possible limitation, subjective steps allow the necessary flexibility to estimate free sugar content of foods when we have less information [46]. Another limitation is that the FLIP database used as reference was built in 2013. It is possible that the composition of commercial foods has changed slightly since then, but it was the most recent database we could use at the time of conducting analyses related to sugar differentiation. However, the fact that we applied FLIP free over total sugars ratio to total sugar content of food items in our database, which are from the 2015 version of the CNF, mitigates this limitation. Finally, given that the %E from free sugars was much lower in under-reporters, but that this was not the case for the %E from naturally occurring sugars, we make the assumption that foods rich in free sugars were more specifically under reported in our sample. To mitigate this limitation, we considered the plausibility of self-reported energy intakes in our analysis.

## 5. Conclusions

The present study is the first to provide data about free and naturally occurring sugar intakes of adults from the Province of Québec using a methodology based on food composition.

The majority of our study participants, no matter the sociodemographic subsample, does not meet WHO’s recommendation to consume less than 10%E from free sugars, which is concerning knowing their associations with a decreased diet quality [47] and weight gain [48], generating potential further detriments to health. In an effort to inform the public and encourage the food industry to change the formulation of their products to reduce the sugar content, governments should consider free sugar labeling on processed foods [49] despite lobbies and various pressures.

## Figures and Tables

**Figure 1 nutrients-11-02317-f001:**
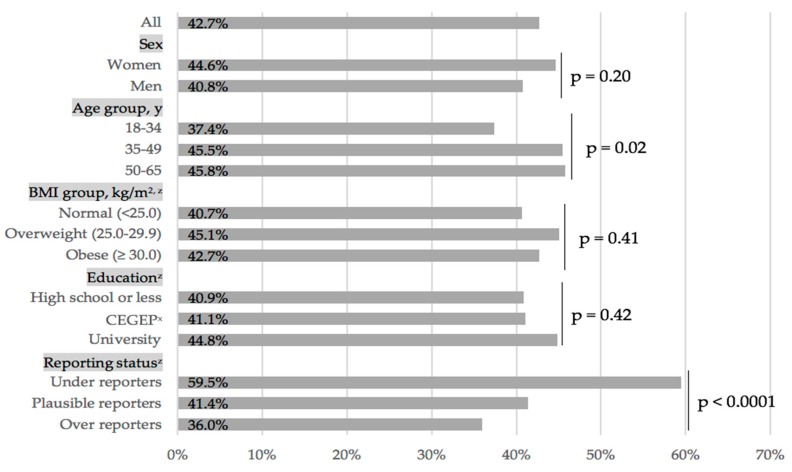
Proportion of individuals meeting the World Health Organization’s recommendation to consume less than 10% of daily kilocalories as free sugars, in a representative sample of French-speaking adults of the Province of Québec, Canada. A *p* value <0.001 was considered statistically significant. ^z^ Missing characteristics were imputed. See the Methods section for details. ^x^ CEGEP is a pre-university and technical college institution specific to the Quebec educational system. BMI, body mass index.

**Table 1 nutrients-11-02317-t001:** Method for calculating the content of free sugars in foods and recipes (adapted from Bernstein et al. [29]).

**Step 1 (as per Step 1 of Bernstein et al. [29]):** Foods that contain no total sugar: By definition, contain 0 g free sugar/100 g. E.g., Butter, meats.	(7.9%)
**Step 2 (as per Step 2 of Bernstein et al. [29]):** Foods that contain no free sugar in the ingredient list: By definition, contain 0 g free sugar/100 g. E.g., Unsweetened fruits, grain products, pasta, eggs, unsweetened milk and yogourt, or milk and yogurt sweetened with non-nutritive sweeteners.	(17.2%)
**Step 3 (as per Step 3 of Bernstein et al. [29]):** Foods that contain no or minimal amounts of naturally occurring sugars: By definition, all total sugars are considered free sugars. E.g., Soft drinks, candies.	(6.5%)
**Step 4 (as per Step 4 of Bernstein et al. [29]):** Foods that contain both naturally occurring and free sugars are compared to similar foods without free sugar ingredients (foods from steps 1 and 2): As much as possible, comparisons are made between sweetened and unsweetened products with similar ingredients or flavors. E.g., chocolate skim milk (10.34 g of total sugars/100 g) vs. unsweetened skim milk (5.09 g of total sugars/100 g)	(6.3%)
$Free\;sugars:\;\frac{100\times \left(\frac{sugars}{100\;g}\;of\;the\;unsweetened\;product-\frac{sugars}{100\;g}\;of\;the\;sweetened\;product\right)}{\frac{sugars}{100\;g}of\;the\;unsweetened\;product-100}$	(1)
$Free\;sugars/100\;g\;in\;chocolate\;skim\;milk:\;\frac{100\times \left(5.09-10.34\right)}{5.09-100}=5.53\;g$	(2)
**Step 5 (adapted from Step 5 of Bernstein et al. [29]):** Foods that contain both naturally occurring sugars and free sugars for which no similar unsweetened food could be used for comparison: The reference database used was the Food Label Information Program (FLIP) 2013, which inventories package label information of foods and beverages of the Canadian food market. commercial foods and beverages [29], determined with Bernstein et al.’s algorithm [29]. A ratio of free sugars over total sugars was calculated for each food in FLIP. When using this step, free sugars are estimated by multiplying the total sugar content of a food in the R24W database by the mean free sugars to total sugars ratio of all similar foods from the FLIP database.FLIP 2013 also contains nutrition information, including total and free sugars, of 15,342	(16.0%)
**Step 6 (as per Step 6 of Bernstein et al. [29]):** Items that could not be compared to similar foods in the FLIP database: Amount of free sugars is determined by using the ratio of free sugars over total sugars of a product in the same food category. E.g., Free sugar content of pouding chômeur, which is a traditional Québec dessert consisting of a cake dough cooked in brown sugar and cream, was determined using the same ratio as sugar pie.	(0.4%)
**Step 7 (new step added):** This step is used for mixed dishes manually created, based on Canadian Nutrient File (CNF) 2015. The ingredients and their quantity are specified for each recipe. Each ingredient is, itself, subjected to the sugar differentiation method (one of the steps above).	(45.6%)
**Step 8 (new step added):** When neither of the steps above could be used to determine free sugar content in foods, an intuitive and logical method is used. E.g., Free sugar content of breakfast powder drink mix was determined by subtracting naturally occurring sugar from powdered milk, estimated based on protein content.	(0.1%)

The percentage of items of PREDISE’s R24W food database identified at each step is indicated within brackets.

**Table 2 nutrients-11-02317-t002:** Mean (95% CI) intakes of total, free and naturally occurring sugars, expressed in grams, percentages of total sugars and percentages of energy they provide, in a representative sample of French-speaking adults of the Province of Québec, Canada.

	*n* (Weighted)	Total Sugars, g	Percentage of Total Sugars, %		Energy, kcal	Percentage of Daily Energy (%E) Provided by, %kcal		
			Free Sugars	Naturally Occurring Sugars		Total Sugars	Free Sugars	Naturally Occurring Sugars
All	1147	116.4 (113.3–119.6)	59.6 (58.6–60.6)	40.4 (39.4–41.4)	2402 (2362–2443)	19.3 (18.9–19.6)	11.7 (11.4–12.1)	7.5 (7.3–7.7)
Sex								
Women	576	97.7 (92.8–102.6) ^a^	57.6 (55.8–59.5) ^a^	42.4 (40.5–44.2) ^a^	2001 (1961–2042) ^a^	19.5 (18.7–20.3)	11.4 (10.8–12.1)	8.1 (7.6–8.5) ^a^
Men	571	124.9 (119.0–130.7) ^b^	63.0 (61.1–65.0) ^b^	37.0 (35.0–38.9) ^b^	2626 (2575–2678) ^b^	18.7 (18.0–19.5)	12.2 (11.5–12.9)	6.5 (6.1–6.9) ^b^
*p*		<0.0001	<0.0001	<0.0001	<0.0001	0.05	0.02	<0.0001
Age group, years								
18–34	408	120.5 (114.1–126.9) ^a^	63.5 (61.4–65.6) ^a^	36.5 (34.4–38.6) ^a^	2458 (2401–2514) ^a^	19.4 (18.6–20.3)	12.6 (11.9–13.4) ^a^	6.8 (6.3–7.3)
35–49	338	111.1 (104.7–117.5) ^b^	59.7 (57.4–62.0) ^b^	40.3 (38.0–42.6) ^b^	2329 (2278–2381) ^b^	18.9 (18.0–19.8)	11.7 (10.8–12.5) ^a,b^	7.2 (6.7–7.7)
50–65	400	102.2 (97.0–107.5)^c^	57.8 (55.7–59.8) ^b^	42.2 (40.2–44.3) ^b^	2154 (2108–2200) ^c^	19.0 (18.2–19.8)	11.2 (10.5–11.9) ^b^	7.8 (7.4–8.3)
*p*		<0.0001	<0.0001	<0.0001	<0.0001	0.47	0.0006	0.001
Administrative region								
Estrie	110	106.3 (97.6–115.1)	59.1 (55.7–62.6)	40.9 (37.4–44.3)	2321 (2257–2385)	18.1 (16.9–19.2)	11.0 (9.9–12.1)	7.0 (6.4–7.7)
Saguenay-Lac-Saint-Jean	107	120.5 (111.0–130.1)	63.1 (60.1–66.2)	36.9 (33.8–39.9)	2341 (2263–2420)	20.4 (19.2–21.7)	13.2 (12.1–14.4)	7.2 (6.6–7.9)
Capitale-Nationale/Chaudière-Appalaches	435	111.0 (105.9–116.1)	59.5 (57.4–61.5)	40.5 (38.5–42.6)	2315 (2271–2360)	19.0 (18.3–19.8)	11.6 (10.9–12.3)	7.4 (6.9–7.9)
Montréal	397	110.5 (105.6–115.4)	58.4 (56.5–60.4)	41.5 (39.6–43.5)	2321 (2275–2368)	19.0 (18.3–19.8)	11.4 (10.7–12.1)	7.6 (7.2–8.1)
Mauricie	99	108.0 (97.1–118.9)	61.5 (58.2–64.8)	38.5 (35.2–41.8)	2269 (2162–2377)	19.1 (17.6–20.6)	11.9 (10.7–13.2)	7.1 (6.3–8.0)
*p*		108.0 (97.1–118.9)	0.06	0.06	0.87	0.07	0.03	0.48
Body mass index, kg/m^2^								
Normal (<25.0)	453	106.1 (100.4–111.8) ^a^	59.3 (57.3–61.4)	40.7 (38.6–42.7)	2146 (2098–2194) ^a^	19.5 (18.7–20.3)	11.9 (11.1–12.6)	7.6 (7.2–8.1)
Overweight (25.0–29.9)	383	106.5 (100.4–112.6) ^a^	59.3 (57.2–61.5)	40.7 (38.5–42.8)	2286 (2231.9–2340) ^a^	18.6 (17.8–19.5)	11.3 (10.5–12.0)	7.4 (6.8–7.9)
Obese (≥30.0)	312	121.3 (115.2–127.4) ^b^	62.3 (60.1–64.4)	37.7 (35.6–39.9)	2509 (2457.7–2561) ^b^	19.2 (18.4–20.1)	12.4 (11.6–13.2)	6.9 (6.4–7.3)
*p*		<0.0001	0.03	0.03	<0.0001	0.10	0.03	0.01
Education								
High school or less	284	113.2 (106.5–120.0)	62.7 (60.5–65.0) ^a^	37.3 (35.0–39.5) ^a^	2329 (2271–2388)	19.1 (18.2–20.1)	12.3 (11.4–13.1)	6.9 (6.3–7.4) ^a^
CEGEPx	353	108.5 (102.9–114.2)	60.9 (58.8–63.1) ^a^	39.1 (36.9–41.2) ^a^	2309 (2258–2361)	18.8 (18.0–19.6)	11.8 (11.0–12.5)	7.0 (6.6–7.5) ^a^
University	510	112.0 (106.1–117.9)	57.3 (55.2–59.4) ^b^	42.7 (40.6–44.8)^b^	2302 (2255–2350)	19.4 (18.6–20.3)	11.5 (10.7–12.2)	7.9 (7.5–8.4) ^b^
*p*		0.32	<0.0001	<0.0001	0.70	0.27	0.22	0.0001
Reporting status								
Under reporters	185	64.0 (58.1–69.8) ^a^	55.1 (52.1–58.2) ^a^	44.9 (41.8–47.9) ^a^	1471 (1414–1528) ^a^	17.9 (16.8–19.1)	10.2 (9.2–11.2) ^a^	7.7 (7.1–8.4)
Plausible reporters	613	109.7 (104.9–114.6) ^b^	61.7 (59.9–63.5) ^b^	38.3 (36.5–40.1) ^b^	2264 (2226.0–2302) ^b^	19.4 (18.7–20.2)	12.3 (11.6–13.0) ^b^	7.1 (6.7–7.5)
Over reporters	348	160.1 (152.5–167.8) ^c^	64.1 (62.1–66.1) ^b^	35.9 (33.9–37.9) ^b^	35.9 (33.9–37.9) ^b^	20.0 (19.2–20.8)	13.0 (12.3–13.7) ^b^	7.0 (6.5–7.5)
*p*		<0.0001	<0.0001	<0.0001	<0.0001	0.005	<0.0001	0.14

Daily intakes are presented as means (95% Confidence Interval) adjusted for sex, age group, administrative region, BMI, education, reporting status, household income level and weekend, when appropriate (except All). *p* Values are the partial effect of sociodemographic characteristics on sugar intake in the linear model. A *p* value <0.001 was considered statistically significant. ^a,b,c^ Subgroups least square means without a common superscript are different (Tukey-Kramer). x CEGEP is a preuniversity and technical college institution specific to the Province of Quebec.

**Table 3 nutrients-11-02317-t003:** Correlations between %E from total, free and naturally occurring sugars in overall diet and in foods and drinks separately in a representative sample of French-speaking adults of the Province of Québec, Canada.

		Total Sugars	Free Sugars	Naturally Occurring Sugars
Overall diet	total sugars	-	0.79 (*p* < 0.0001)	0.45 (*p* < 0.0001)
	free sugars	-	-	−0.20 (*p* < 0.0001)
Foods	total sugars	-	0.69 (*p* < 0.0001)	0.66 (*p* < 0.0001)
	free sugars	-	-	−0.09 (*p* = 0.01)
Drinks	total sugars	-	0.91 (*p* < 0.0001)	0.39 (*p* < 0.0001)
	free sugars	-	-	−0.02 (*p* = 0.61)

A *p* value <0.001 was considered statistically significant.

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
