# Peer review of "Intakes of Total, Free, and Naturally Occurring Sugars in the French-Speaking Adult Population of the Province of Québec, Canada: The PREDISE Study"

_nutrients, 2019, doi:10.3390/nu11102317_

Round 1
Reviewer 1 Report
This paper describes the intakes of total, free and naturally occurring sugars in the French-speaking adult population in Québec, Canada, using cross-sectional data from the PREDISE study. This is an excellent paper with interesting and well discussed surveillance data. The paper is well presented and addresses a hot topic in nutritional epidemiology. The authors add important findings to this field and provide valuable methodology on population intakes of these 3 carbohydrate types.
Some minor edits/suggestions:
Chapter “Participants and procedures”: Please add the year(s) of data collection. Line 168: The formula is unclear (especially the bottom). Please clarify or provide an example with numbers. Methodologically speaking, it is common to model usual intake out of the three 24-hour dietary recalls with software, such as MSM or NCI. Could you comment on your choice of not doing so? Line 227: Considering the number of statistical tests you used, you should apply correction for multiple testing. In other words, you should set a significance level for P at 0.001 (or below) instead of 0.05. Consider rephrasing some of your results accordingly. Information about main food sources of free and naturally occurring sugars (not just foods vs. drinks) would be a valuable input for your article. Why did not you include this part in your article? Title of Table 2: Please write “Table 2. Mean (95% CI) intake of total …” to facilitate reading. Lines 238-256: This paragraph is hard to read. Please consider following the same order between the table and the text to make reading easier. Lines 378-381 and 411-413: Your conclusion highlights the importance of considering free sugar labelling on processed foods instead of total sugar, but the discussion is not fully clear about it. Please provide more information in the discussion to be able to conclude in such a way. This will be particularly necessary for foods (less for drinks, based on your results).
Author Response
Please see the attachment
The abstract was revised in the manuscript and a new version of the Supplemental file has been uploaded.

Reviewer 2 Report
The present study is the first to provide data about free and naturally occurring sugar intakes of adults from the Province of Québec using a methodology based on food composition. This paper is generally well written and structured. However, I would suggest the author revise some places to help readers’ understand.
#minor points
1) There is a description of the household income levels in Canadian dollars (<30,000, ≥30,000 to <60,000, ≥60,000 to <90,000, ≥90,000) and the number of 24-hour dietary recalls completed on week-end days (0, 1, 2, 3) in the method section and in the footnote of Table 2, , but Table 2 and Figure 1 do not have results. Is there a special reason for that? If not, I would suggest showing this result in addition.
2) The author described the correlations among types of sugars in the results section (Line 275 ~). If you present these results as a table, I think it will be easier for the reader to understand them.
Author Response

(The authors gave the same response as above.)

Reviewer 3 Report
The manuscript deals with an interesting topic and requests minor revisions.
In the algorithm that you proposed at step 3 foods with no naturally occurring sugars are included. Most of these foods could contain intense sweeteners rather than added sugar and thus how did you consider in your method?
Statistical analyses:
Should be explain why the multivariate linear regression model was adopted to obtain the means of dietary intakes.
Results:
It would be interesting to add a description of the main sources of free sugars in terms of food categories and possibly sub-categories.
Discussion
Row 306: please specify that “other” food is one of the food categories used in the Canadian categorisation system otherwise the sentence is not clear.
Author Response

(The authors gave the same response as above.)
